# UNSUPERVISED DEMIXING OF STRUCTURED SIGNALS FROM THEIR SUPERPOSITION USING GANS

## ABSTRACT

Recently, Generative Adversarial Networks (GANs) have emerged as a popular alternative for modeling complex high dimensional distributions. Most of the existing works implicitly assume that the clean samples from the target distribution are easily available. However, in many applications, this assumption is violated. In this paper, we consider the observation setting in which the samples from a target distribution are given by the superposition of two structured components, and leverage GANs for learning of the structure of the components. We propose a novel framework, *demixing-GAN*, which learns the distribution of two components at the same time. Through extensive numerical experiments, we demonstrate that the proposed framework can generate clean samples from unknown distributions, which further can be used in demixing of the unseen test images.

## 1 INTRODUCTION

In this paper, we consider the classical problem of separating two structured signals observed under the following superposition model:

$$Y = X + N, \tag{1}$$

where $X \in \mathcal{X}$ and $N \in \mathcal{N}$ are the *constituent signals/components*, and $\mathcal{X}, \mathcal{N} \subseteq \mathbb{R}^p$ denote the two structured sets. In general the separation problem is inherently ill-posed; however, with enough structural assumption on $\mathcal{X}$ and $\mathcal{N}$, it has been established that separation is possible. Depending on the application one might be interested in estimating only $X$ (in this case, $N$ is considered as the corruption), which is referred to as *denoising*, or in recovering (estimating) both $X$ and $N$ which is referred to as *demixing*. Both denoising and demixing arise in a variety of important practical applications in the areas of signal/image processing, computer vision, machine learning, and statistics (Chen et al., 2001; Elad et al., 2005; Bobin et al., 2007; Candès et al., 2011). Most of the existing demixing techniques assume some prior knowledge on the structures of $\mathcal{X}$ and $\mathcal{N}$ in order to recover the desired component signals. Prior knowledge about the structure of $\mathcal{X}$ and $\mathcal{N}$ can only be obtained if one has access to the generative mechanism of the signals and clean samples from the probability distribution defined over sets $\mathcal{X}$ and $\mathcal{N}$. In many practical settings, none of these may be feasible.

In this paper, we consider the problem of separating constituent signals from superposed observations when we do not have access to the clean samples from any of the distributions (fully unsupervised approach). In particular, we are given a set of superposed observations $\{Y_i = X_i + N_i\}_{i=1}^K$ where $X_i \in \mathcal{X}$ and $Y_i \in \mathcal{N}$ are i.i.d samples from their respective (unknowns) distributions. In this setup, we explore two questions: First, *How can one learn the prior knowledge about the individual components from superposition samples?*; hence, we concern with a learning problem. Second, *Can we leverage the implicitly learned constituent distributions for tasks such as demixing of a test image?*. As a result, in the latter question, we deal with a inference task.

### 1.1 SETUP AND OUR TECHNIQUE

Motivated by the recent success of generative models in high-dimensional statistical inference tasks such as compressed sensing in Bora et al. (2017; 2018), in this paper, we focus on Generative Adversarial Network (GAN) based generative models to implicitly learn an unknown distribution, i.e., generate samples from the learned distribution. Most of the existing works on GANs typically assume

access to clean samples from the underlying signal distribution. However, this assumption clearly breaks down in the superposition model considered in our setup, where the structured superposition makes training generative models very challenging. Our contribution in this paper is to answer the above questions. Specifically, we address the first question (learning problem) by proposing a novel GAN framework, which we call it *demixing*-GAN. In particular, we show through extensive empirical simulations over three different datasets that the demixing-GAN can learn the unknown distribution of the constituent components by generating the samples from each of them. To answer the second question (inference problem), we use our trained generators from the demixing-GAN to demix the unseen test samples (i.e., samples not used in the training of the demixing-GAN) by discovering the best hidden representation of the constituent components from the generative models.

The rest of this paper is organized as follows: In section 2, we discuss relevant previous works and compare our novelty over these existing methods. In section 3, we formally introduce our proposed approach, and in section 4, we provide extensive experimental results to validate our framework. Finally, we conclude this paper by providing a summary in section 5.

## 2 PRIOR ART

To overcome the inherent ambiguity issue in problem (1), many existing methods have assumed that the structure of sets $\mathcal{X}$ and $\mathcal{N}$ (i.e., the structures can be low-rank matrices, or have sparse representation in some domain (McCoy & Tropp, 2014)) is a prior known and also that the signals from $\mathcal{X}$ and $\mathcal{N}$ are "distinguishable" (Elad & Aharon, 2006; Soltani & Hegde, 2016; 2017; Druce et al., 2016; Elyaderani et al., 2017; Jain et al., 2017). Knowledge of the structures is a big restriction in many real-world applications. Recently, there have been some attempts to automate this *hard-coding* approach. Among them, structured sparsity (Hegde et al., 2015), dictionary learning (Elad & Aharon, 2006), and in general manifold learning are the prominent ones. While these approaches have been successful to some extent, they still cannot fully address the need for the prior structure. Over the last decade, deep neural networks have been demonstrated that they can learn useful representations of real-world signals such as natural images, and thus have helped us to understand the structure of the high dimensional signals, for e.g., using deep generative models (Ulyanov et al., 2017). In this paper, we focus on Generative Adversarial Networks GANs) (Goodfellow et al., 2014) as the generative models for implicitly learning the distribution of constituent components. GANs have been established as a very successful tool for generating structured high-dimensional signals (Berthelot et al., 2017; Vondrick et al., 2016) as they do not directly learn a probability distribution; instead, they generate samples from the target distribution(s) (Goodfellow, 2016). In particular, if we assume that the structured signals are drawn from a distribution lying on a low-dimensional manifold, GANs generate points in the high-dimensional space that resemble those coming from the true underlying distribution.

Since their inception, there has been a flurry of works on GANs (Zhu et al., 2017; Yeh et al., 2016; Subakan & Smaragdis, 2018) to name a few. In most of the existing works on GANs with a few notable exceptions such as Wu et al. (2016); Bora et al. (2018); Kabkab et al. (2018); Hand et al. (2018); Zhu et al. (2016), it is implicitly assumed that one has access to clean samples of the desired signal. However, in many practical scenarios, the desired signal is often accompanied by unnecessary components.

Recently, GANs have also been used for capturing the structure of high-dimensional signals specifically for solving inverse problems such as sparse recovery, compressive sensing, and phase retrieval (Bora et al., 2017; Kabkab et al., 2018; Hand et al., 2018). For instance, Bora et al. (2017) has shown that generative models provide a good prior for structured signals, for e.g., natural images, under compressive sensing settings over sparsity-based recovery methods. They rigorously analyze the statistical properties of a generative model based on compressed sensing and provide some theoretical guarantees and experimental evidence to support their claims. However, they don't explicitly propose an optimization procedure to solve the recovery problem. They simply suggest using stochastic gradient-based methods in the low-dimensional latent space to recover the signal of interest. This has been addressed in Shah & Hegde (2018), where the authors propose using a projected gradient descent algorithm for solving the recovery problem directly in the ambient space (space of the desired signal). They provide theoretical guarantees for the convergence of their algorithm and also demonstrate improved empirical results over Bora et al. (2017).

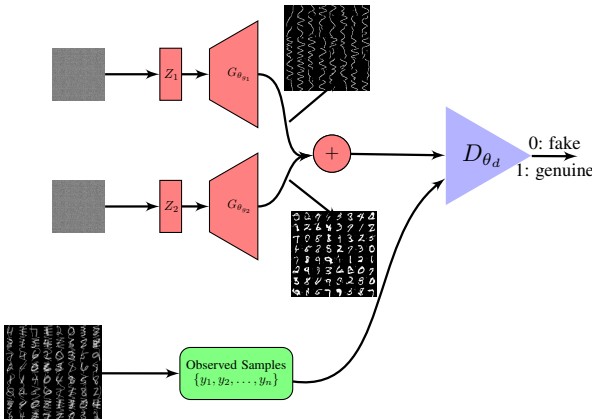

Figure 1: The architecture of the proposed demixing-GAN.

While GANs have found many applications, most of them need direct access to the clean samples from the unknown distribution, which is not the case in many real applications such as medical imaging. AmbientGAN framework Bora et al. (2018) partially addresses this problem. In particular, they studied various measurement models and showed that their GAN model can find samples of clean signals from corrupted observations. However, AmbientGAN assumes that the observation model and parameters are known. That is, they have assumed that one has access to the samples of corruption part, which is a strong restriction in the real-world applications. One of our main contributions is addressing this limitation by studying the demixing problem. Actually, if we can learn the distribution of both components (e.g., generating samples from them), then we can use the samples of the second component (corruption part) for downstream task such as denoising without explicitly needing the samples from the corruption part. That is why our framework is a purely unsupervised approach. In addition, AmbientGAN just learns the distribution of the clean images; however, it has not been used for the task of image denoisng (i.e., how to denoise an unseen corrupted image). Our framework addresses this issue in a general scenario of demixing of unseen test images.

## 3    BACKGROUND AND PROPOSED IDEA

### 3.1    PRELIMINARIES

Generative Adversarial Networks (GANs) are one of the successful generative models in practice was first introduced by Goodfellow et al. (2014) for generating samples from an unknown target distribution. As opposed to the other approaches for density estimation such as *Variational Auto-Encoders (VAEs)* Kingma & Welling (2013), which try to learn the distribution itself, GANs are designed to generate samples from the target probability density function. This is done through a zero-sum game between two players, *generator*, $G$ and *discriminator*, $D$ in which the generator $G$ plays the role of producing the fake samples and discriminator $D$ plays the role of a cop to find the fake and genuine samples. Mathematically, this is accomplished through the following *min-max* optimization problem:

$$\min_{\theta_g} \max_{\theta_d} \quad \mathbb{E}_{x \sim \mathcal{D}_x}[log(D_{\theta_d}(x))]\mathbb{E}_{z \sim \mathcal{D}_z}[log(1 - D_{\theta_d}(G_{\theta_g}(z)))], \tag{2}$$

where $\theta_g$ and $\theta_d$ are the parameters of generator network and discriminator network, respectively. $\mathcal{D}_x$ denotes the target probability distribution, and $\mathcal{D}_z$ represents the probability distribution of the hidden variables $z \in \mathbb{R}^h$, which is assumed either a uniform distribution in $[-1, 1]^h$, or standard normal. One can also use identity function instead of $log(.)$ function in the above expression. The resulting formulation is called WGAN A. et al. (2017). It has been shown that if $G$ and $D$ have enough capacity, then solving optimization problem (2) by alternative stochastic gradient descent guarantees the distribution $\mathcal{D}_g$ at the output of the generator converges to $\mathcal{D}_x$. Having discussed the basic setup of GANs, next we present the proposed modifications to the basic GAN setup that allows for usage of GANs as a generative model for demixing structured signals.

## 3.2 DEMIXING-GAN

In this section, we discuss our main contribution. First, we start with learning problem (the first question in the introduction section) and explain our GAN framework for learning the distribution of two constituent components. Next, we move to the inference part (the second question in the introduction section) to show that how we can leverage the learned generating process for demixing of a test mixed image. Finally, we provide some theoretical intuition about the success/failure of the demixing-GAN.

### 3.2.1 LEARNING

Figure 1 shows the GAN architecture, we use for the purpose of separating or demixing of two structured signals form their superposition. As illustrated, we have used two generators and have fed them with two random noise vectors $z_1 \in \mathbb{R}^{h_1}$ and $z_2 \in \mathbb{R}^{h_2}$ according to a uniform distribution defined on a hyper-cube, where $h_1, h_2$ are less than the dimension of the input images. We also assume that they are independent of each other. Next, the output of generators are summed up and the result is fed to the discriminator along with the superposition samples, $y_i's$. In Figure 1, we just show the output of each generator after training for an experiment case in which the mixed images consist of 64 MNIST binary image (LeCun & Cortes, 2010) (for $X$ part) and a second component constructed by random sinusoidal (for $N$ part) (please see the experiment section for more details). Somewhat surprisingly, the architecture based on two generators can produce samples from the distribution of each component after enough number of training iterations. We note that this approach is fully unsupervised as we only have access to the mixed samples and nothing from the samples of constituent components is known. As mentioned above, this is in sharp contrast with the AmbientGAN. As a result, the demixing-GAN framework can generate samples from both components (If for example the second component is drawn from a random sinusoidal, then the generated samples can be used in the task of denoising where the corruption components are sampled from highly structured sinusoidal waves).

### 3.2.2 INFERENCE

Now, we can use the trained generators in Figure 1 for demixing of the constituent components for a given test mixed image which has not been used in training. To this end, we use our assumption that the components have some structure and the representation of this structure is given by the last layer of the trained generator. This observation together with this fact that in GANs, the low-dimension random vector $z$ is representing the hidden variables, leads us to this point: in order to demix a new test mixed image, we have to find a hidden representations corresponding to each component which give the smallest distance to the constituent images in the space of $G_{\widehat{\theta}_{g_1}}$ and $G_{\widehat{\theta}_{g_2}}$ (Shah & Hegde, 2018; Bora et al., 2017). In other words, we have to solve the following optimization problem:

$$\widehat{z_1}, \widehat{z_2} = \arg\min_{z_1, z_2} \|y - G_{\widehat{\theta}_{g_1}}(z_1) - G_{\widehat{\theta}_{g_2}}(z_2)\|_2^2 + \lambda_1\|z_1\|_2^2 + +\lambda_2\|z_2\|_2^2, \tag{3}$$

where $u$ denotes the test mixed image. Now, each component can be estimated by evaluating $G_{\widehat{\theta}_{g_1}}(\widehat{z_1})$ and $G_{\widehat{\theta}_{g_2}}(\widehat{z_2})$[1]. While the optimization problem in (3) is non-convex, we can still solve it through an alternative minimization fashion. We note that in optimization problem (3), we did not project on the box sets on which $z_1$ and $z_2$ lie on. Instead we have used regularizer terms in the objective functions (which are not meant as projection step). We empirically have observed that imposing these regularizers can help to obtain good quality images in our experiment; plus, they may help that the gradient flow to be close in the region of interest by generators. This is also used in Bora et al. (2017). Now, we provided some theoretical intuitions for the demixing-GAN.

### 3.2.3 SOME THEORETICAL INTUITIONS ABOUT THE DEMIXING-GAN

Recall that the superposition model is given by $Y = X + N$, and $\mathcal{D}_y$, $\mathcal{D}_x$ and $\mathcal{D}_n$ denote the distribution of $Y, X$, and $N$, respectively. Let $G_1(z_1) \triangleq G_{\theta_{g_1}}(z_1) \sim \mathcal{D}_{g_1}$ and $G_2(z_2) \triangleq G_{\theta_{g_2}}(z_2) \sim \mathcal{D}_{g_2}$. Also assume that $(z_1, z_2) \sim \mathcal{D}_{z_1, z_2}$ denotes the joint distribution of the hidden random vectors

---

[1]$G_{\widehat{\theta}_{g_1}}(.)$ and $G_{\widehat{\theta}_{g_2}}(.)$ denote the first and second trained generator with parameter $\widehat{\theta}_{g_1}$ and $\widehat{\theta}_{g_2}$, respectively.

with marginal probability as $\mathcal{D}_{z_i}$ for $i = 1, 2$. We note that in demxing setting there are not samples from the component $N$ as opposed to the typical denoising scenarios. Now we have the following mini-max loss as (2):

$$\min_{G_1, G_2} \max_{D} \mathcal{L}(G_1, G_2, D) = \mathbb{E}_{u \sim \mathcal{D}_y} log(D(u)) + \mathbb{E}_{(z_1, z_2) \sim \mathcal{D}_{z_1, z_2}} log(1 - D(G_1(z_1) + G(z_2))).$$

Following the standard GAN framework, for the fixed $G_1$ and $G_2$, we have:

$$\mathcal{L}(G_1, G_2, D) = \int_u (\mathcal{D}_y(u) log(D(u)) + \mathcal{D}_G(u) log(1 - D(u))) du,$$

where $\mathcal{D}_G = \mathcal{D}_{g_1} * \mathcal{D}_{g_2}$. Hence, the optimal discriminator is given by $D^* = \frac{\mathcal{D}_x * \mathcal{D}_n}{\mathcal{D}_x * \mathcal{D}_n + \mathcal{D}_{g_1} * \mathcal{D}_{g_2}}$ since $Y = X + N$ and the fact that $\mathcal{D}_y$ and $\mathcal{D}_g$ are the pdf and defined over $[0, 1]$. This means that the global optimal of the above optimization problem is achieved iff $\mathcal{D}_x * \mathcal{D}_n = \mathcal{D}_{g_1} * \mathcal{D}_{g_2}$ ( $*$ denotes the convolution operator). However, this condition is generally an ill-posed equation. That is, in general, $\mathcal{D}_x \neq \mathcal{D}_{g_1}$ and $\mathcal{D}_n \neq \mathcal{D}_{g_2}$. In the best case, we can have hope to uniquely determine the distributions up to a permutation (similar thing is also true for the ICA method). This is the point actually we need some notion of incoherence between two constituent structures, $\mathcal{X}$, and $\mathcal{N}$. So, the question is this under what incoherent condition, we can have a well-conditioned equation? Also, by using the above optimality condition and taking the Fourier transform (or using the characteristic function, $\Phi(.)$) from this optimality equation, we obtain: $\Phi_{g_1}.\Phi_{g_2} = \Phi_x.\Phi_n$. As a result, $\mathcal{D}_x = \Phi_x^{-1}(\frac{\Phi_{g_1}.\Phi_{g_2}}{\Phi_n})$. For $\mathcal{D}_x = \Phi_x^{-1}(\frac{\Phi_{g_1}.\Phi_{g_2}}{\Phi_n})$ to be well-defined, a straightforward condition is that the $\Phi_n$ should be non-zero almost everywhere. As a result, even if we somehow figure out the right incoherent condition, $\mathcal{D}_x$ is uniquely determined by $\mathcal{D}_n$ if the Fourier transform of $\mathcal{D}_n$ is non-zero. While we currently do not have a right answer for the above question, we conjecture that in addition to the incoherence issue in the signal domain, the hidden space ($z$-space) in both generators play an important role to make the demixing problem possible. We investigate this idea and the other things empirically in the experiment section.

## 4 NUMERICAL EXPERIMENTS

In this section, we present various experiments showing the efficacy of the proposed framework (depicted in Figure 1) in two different setups. First, we will focus on learning the structured distributions from the superposition observation samples. Next, we explore the use of generative models from the proposed GAN framework in an inference task. In all the following experiments, we did our best for choosing all the hyper-parameters.

### 4.1 DEMIXING OF STRUCTURED SIGNALS – TRAINING

In this section, we present the results of our experiments for learning the distribution of the constituent components on different datasets. We first present our experiment with MNIST dataset, and then we show the similar set of experiments with Fashion-MNIST dataset (F-MNIST) Han et al. (2017). Next, we illustrate the performance of the demixing-GAN on Quick-Draw Qui (a). Finally, we present some experimental investigation for the conditions under which the demixing-GAN is failed.

### 4.1.1 EXPERIMENTS ON MNIST DATASET

We start the experiments with considering four sets of constituent components. We have used the network architectures for discriminator and generators similar to the one proposed in DCGAN (Radford et al., 2015). DCGAN is a CNN based-GAN consists of convolutional layers followed by batch normalization (except the last layer of the generator and first layer of discriminator). In the first two experiments in this section, we use the superposition of the MNIST digits and some random corruption signals. That is, we mix the MNIST digits with two (structured) corruption signals: random sinusoidal waves, and random vertical and horizontal lines. In particular, first we generate random sinusoidal waves in which the amplitude, frequency, and phase are random numbers, and second we construct the random vertical and horizontal lines. In Figure 2, we show the training evolution of two fixed random vectors, $z_1$ and $z_2$ in $\mathbb{R}^{100}$ in the output of two generators. In the top left panel, we have added one random sinusoidal wave to the clean digits. As we can see, our

proposed GAN architecture can learn two distributions and generate samples from each of them. In the bottom left panel, we repeat the same experiment with random vertical and horizontal lines as the second component (two random vertical and two random horizontal lines are added to the clean digits). While there is some notion of mode collapse, still two generators can produce the samples from the distribution of the constituent components.

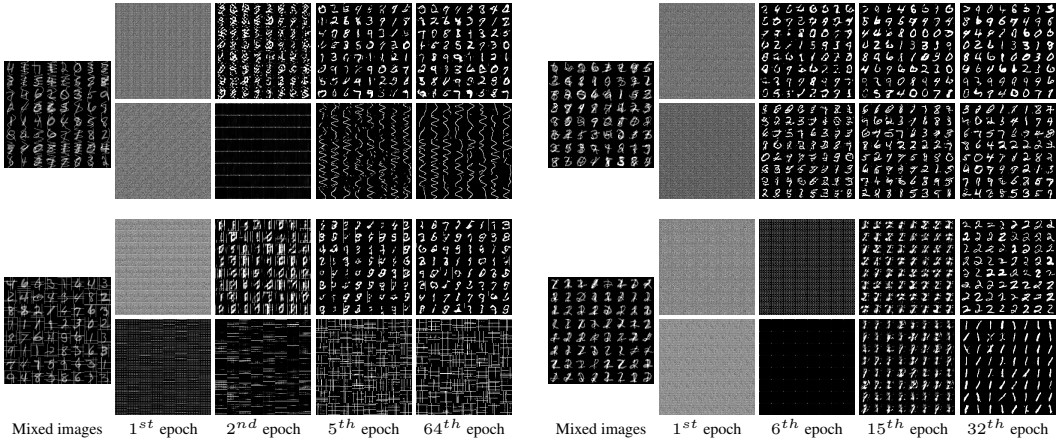

Figure 2: Evolution of output samples by two generators for fixed $z_1$ and $z_2$. The left panel shows the evolution of two generators in different epochs where the mixed images comprise of digits, and sinusoidal (top) and lines (bottom). The first generator is learning the distribution of MNIST digits, while the second one is learning the structured corruptions. The right panel shows the similar experiment with MNIST digits as constituent components. The top mixes all the digits, while the bottom mixes only digits 1 and 2.

For the third experiment in this section, our mixed images comprise of two MNIST digits from 0 to 9. In this case, we are interested in learning the distribution from which each of the digits is drawn. Similar to the previous cases, the top right panel in Figure 2 shows the evolution of two fixed random vectors, $z_1$ and $z_2$. As we can see, after 32 epochs, the output of the generators would be the samples of MNIST digits. Finally, in the last experiment of this section, we generate the mixed images as the superposition of digits 1 and 2. In the training set of MNIST dataset, there are around 6000 samples from each digit of 1 and 2. We have used these digits to form the set of superposition images. The bottom right panel of Figure 2 shows the output of two generators, which can learn the distribution of two digits. The interesting point in these experiments is that each GAN can learn the different variety of existing digits in MNIST training dataset, and we typically do not see mode collapse, which is a major problem in the training of GANs (Goodfellow, 2016).

### 4.1.2 EXPERIMENTS ON F-MNIST DATASET

In this section, we illustrate the performance of the proposed demixing-GAN for F-MNIST dataset. The training dataset in F-MNIST includes 60000 gray-scale images with size of $28 \times 28$ classified in 10 classes. The different labels denote objects, which include T-shirt/top, Trouser, Pullover, Dress, Coat, Sandal, Shirt, Sneaker, Bag, and Ankle boot. Similar to the experiments with MNIST dataset being illustrated in Figure 2, we train the demixing-GAN where we have used InfoGAN (Chen et al., 2016) architecture for the generators. The architecture of the generators in InfoGAN is very similar to the DCGAN discussed above with the same initialization procedure. The dimension of input noise to the generators is set to 62. We have also used the same discriminator in DCGAN. Left panel in Figure 3 shows the output of two generators, which can learn the distribution of dress and bag images during 21 epochs from mixed dress and bag images.

| | MSE ($1^{st}$ Part) | MSE ($2^{nd}$ Part) | PSNR ($1^{st}$ Part) | PSNR ($2^{nd}$ Part) |
|---|---|---|---|---|
| First Row | 0.04715 | 0.03444 | 13.26476 | 14.62877 |
| Second Row | 0.04430 | 0.03967 | 13.53605 | 14.01344 |
| Third Row | 0.05658 | 0.05120 | 12.47313 | 12.90715 |
| Forth Row | 0.08948 | 0.10203 | 10.48249 | 9.91264 |

Table 1: Numerical Evaluation of the results in the right panel of Figure 3 according to the *Mean Square Error (MSE)* and *Peak Signal-to-Noise ratio (PSNR)* criteria between the corresponding components in the superposition model.

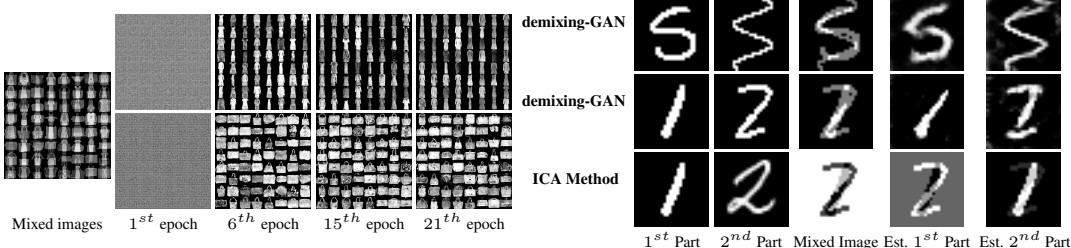

Figure 3: Left Panel: evolution of output samples by two generators for fixed $z_1$ and $z_2$. The mixed images comprise only two objects, dress, and bag in the training F-MNIST dataset. One generator produces the samples from dress distribution, while the other one outputs the samples from the bag distribution. Right panel: the performance of the trained generators for demixing of two constituent components (inference). The first two columns are the ground-truth components. The third column is the ground-truth mixed image and the last two columns denote the recovered components. The first row uses the same generator trained for only one digit (drawn from MNIST test dataset) and a random sinusoidal. The second row uses the generator trained only for digits 1 and 2. The last row shows the result of demixing with ICA method.

## 4.2 DEMIXING OF STRUCTURED SIGNALS – TESTING

We now test the performance of two trained generators in a demixing scenario for the test mixed images, which have not been seen in the training time. Right panel in Figure 3 shows our experiment in which we have illustrated the demixing on three different test mixed images. Here, we have compared the performance of the demixing-GAN with *Independent component analysis (ICA)* method (Hoyer et al., 1999) (We have implemented ICA using Scikit-learn module (Pedregosa et al., 2011)). In the top and middle rows of Figure 3 (right panel), we consider the mixed images generated by adding a digit (drawn from MNIST test dataset) and a random sinusoidal. Then the goal is to separate (demix) these two from their given sum. To do this, we use the GAN trained for learning the distribution of digits and sinusoidal waves (the top left panel of Figure 2) and solve the optimization problem in (3) through an alternative minimization approach. As a result, we obtain $\widehat{z_1}$ and $\widehat{z_2}$. The corresponding constituent components is then obtained by evaluating $G_{\widehat{\theta}_{g_1}}(\widehat{z_1})$ and $G_{\widehat{\theta}_{g_2}}(\widehat{z_2})$. In the right panel of Figure 3, the first two columns denote the ground-truth of the constituent components. The middle one is the mixed ground-truth, and the last two show the recovered components using demixing-GAN and ICA. In the last row, digits 1 and 2 drawn from the MNIST test dataset, are added to each other and we apply the GAN trained for learning the distribution of digits 1 and 2 (bottom right panel in Figure 2). As we can see, our proposed GAN can separate two digits; however, ICA method fails in demixing of two components. In addition, Table 1 has compared numerically the quality of recovered components with the corresponding ground-truth ones through *mean square error (MSE)* and *Peak Signal-to-Noise Ratio (PSNR)* criteria.

Now, we evaluate the performance of the trained demixing-GAN on the F-MNIST dataset. For each panel in Figure 4, the first two columns denote two objects from F-MNIST test dataset as the ground-truth components. The third column is the ground-truth mixed image, and the last two columns show the recovered constituent components similar to the previous case. The top left uses the generator trained for only two objects for 20 epochs. The top right uses the generator trained for all 10 objects for 20 epochs. The bottom left uses the same generator trained for only two objects for 30 epochs. The bottom right shows the result of demixing with ICA method. As we can see, ICA fails to separate the components (images of F-MNIST) from each other, while the proposed demixing-GAN can separate the mixed images from each other. While the estimated image components are not exactly matched to the ground-truth ones (first two columns), they are semantically similar to the ground-truth components.

## 4.3 EXPERIMENTS ON BOTH MNIST AND F-MNIST DATASET

In this section, we explore the performance of the demixing-GAN when the superposed images is the sum of digits 8 from MNIST dataset and dresses from the F-MNIST dataset. The experiment for this setup has been illustrated in the left panel of Figure 5. Since our goal is to separate dress from the digit 8, for the first generator, we have used the InfoGAN architecture being used in the experiment

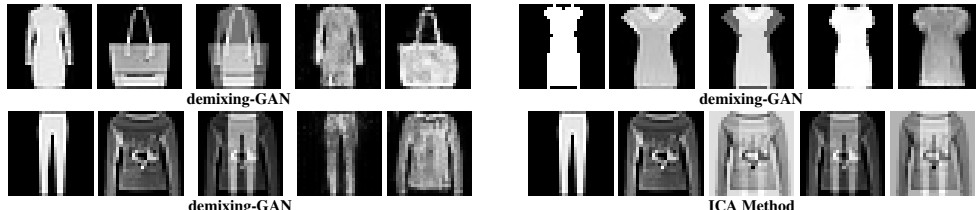

Figure 4: Performance of the trained generators for demixing of two constituent components. In all panels, the first two columns are the ground-truth components. The third column is the ground-truth mixed image and the last two columns denote the recovered components.The top left uses the generator trained for only two objects for 20 epochs. The top right uses the generator trained for all 10 objects for 20 epochs. The bottom left uses the same generator trained for only two objects for 30 epochs. The bottom right shows the result of demixing with ICA method.

in section 4.1.2 and similarly the DCGAN architecture for the second generator as section 4.1.1. As a result, the input noise to the first generator is drawn uniformly from $[-1, 1]^{62}$, and uniformly from $[-1, 1]^{100}$ for the second generator. The left panel of Figure 5 shows the evolution of output samples by two generators for fixed $z_1$ and $z_2$. As we can see, after 21 epoch, the first generator is able to generate dress samples and the second one outputs samples of digit 8.

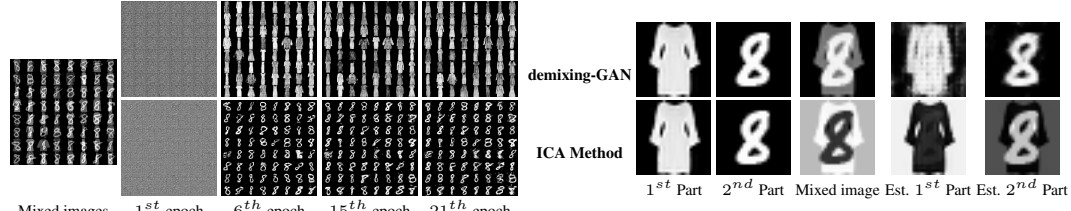

Figure 5: Left Panel: evolution of output samples by two generators for fixed $z_1$ and $z_2$. The mixed images comprise dress from F-MNIST training dataset, and digit 8 from MNIST training dataset. One generator produces the samples from the dress distribution, while the other one outputs the samples from digit 8 distribution. Right Panel: the performance of trained generators for demixing of two constituent components. The first two columns are the ground-truth components. The third column is the ground-truth mixed image and the last two columns denote the recovered components. The first row uses the generator trained through demixing-GAN. The second row shows the result of demixing with ICA method.

Similar to the previous testing scenarios, we now evaluate the performance of the demixing-GAN in comparison with ICA for separating a test image which is a superposition of a digit 8 drawn randomly from MNIST test dataset and dress object drawn randomly from F-MNIST test dataset. Right panel in Figure 5 shows the performance of the demixng-GAN and ICA method. As we can see, ICA fails to demix two images from their superposition, whereas the demixing-GAN is able to separate digit 8 very well and to some extend the dress object from the input superposed image. MSE and PSNR values for the first component using ICA recovery method is given by 0.40364 and 3.94005, respectively. Also, MSE and PSNR for the first component using ICA recovery method is given by 0.15866 and 7.99536, respectively.

## 4.4 EXPERIMENT ON QUICK-DRAW DATASET

In this section, we present our demixing framework for another dataset, Quick-Draw dataset (Qui, a) released by Google recently. The Quick Draw Dataset is a collection of 50 million drawings categorized in 345 classes, contributed by players of the game Quick, Draw! Qui (b). For the the experiment in the left panel of Figure 6, we consider only two objects, face, and flower in the Quick Draw Dataset (training set includes 16000 images of size $28 \times 28$ for each class). As a result, the input mixed images are the superposition of different faces and flowers. The left panel of Figure 6 shows the evolution of the random vectors $z_1$ and $z_2$ (drawn uniformly from $[-1, 1]^{64}$). As we can see, after 31 epochs, one generator can produce various kind of faces, while the other one generates different shapes of flowers.

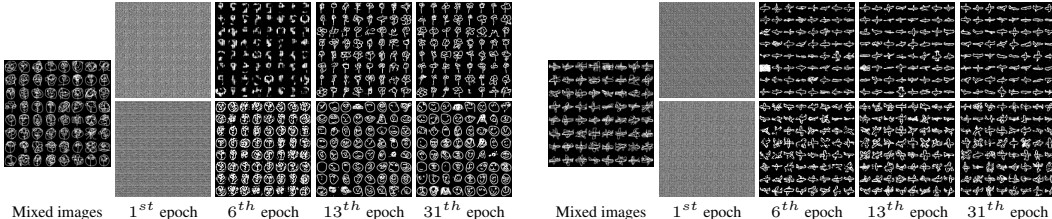

Figure 6: Evolution of output samples by two generators for fixed $z_1$ and $z_2$. Left Panel: the mixed images comprise only two objects, face, and flower in training Quick-Draw dataset. One generator produces the samples from flower distribution, while the other one outputs the samples from the face distribution. Right Panel: the mixed images comprise only airplane object, randomly drawn from the training Quick-Draw dataset. The top generator produces mostly the samples from simpler and flat airplanes, while the bottom one outputs the samples from the more detailed airplane shapes.

Now, we consider a more challenging scenario in which the constituent components in the mixed images are just airplane shapes. That is, we randomly select the airplane shapes from 16000 images in the training set, and add them together to construct the input mixed images. We have been noticed that in the 16000 images of the airplane shapes, in general, there are two structures. One is related to the airplanes having been drawn by the players in more simple and somehow flat manner (they are mostly similar to an ellipse with or without wings) in the Quick, Draw game, while the second one consists of the more detailed shapes (they have a tail and maybe with different orientation). Right panel of Figure 6 depicts the performance of the demixing-GAN for this setup. One surprising point is that while both components in the superposition are drawn from one class (e.g., airplane shapes), the demixing-GAN is still able to demix the hidden structure in the airplane distribution. Thus, we think that just having the same distribution for both of the constituent components is not necessarily a barrier for demixing performance. We conjecture that somehow different features of the shapes drawn from the same distribution makes demixing possible by forcing the enouth incoherence between the components. As we can see, after 31 epochs, both generators can learn two mentioned structures, and regarding two structures, they can cluster the shape of airplanes into two types.

## 4.5 FAILURE OF THE DEMIXING-GAN

In this section, we empirically explore our observation about the failure of the demixing-GAN. We focus on two spaces, hidden space ($z$-space) and signal or generator space (the output of generators) in discovering the failure of demixing-GAN.

Our first observation concerns the $z$-space. We observe that if the hidden vectors form $z$-space of two generators are aligned to each other, then the two generators cannot output the samples in the signal space, representing the distribution of the constituent components. To be more precise, in the left panel of Figure 7, we consider separating digits 8 and 2 from their superpositions. However, here, we feed both generators with the same vector, i.e., $z_1 = z_2$ in each batch (this is considered as the extreme case where precisely the hidden variables equal to each other) and track the evolution of the output samples generated by both generators. As we can see, even after 21 epochs, the generated samples by both generators are an unclear combination of both digits 2 and 8, and they are not separated clearly as opposed to the case when we feed the generators with $i.i.d$ random vectors. We also repeat the same experiment with two aligned vectors $z_1$ and $z_2$, i.e., $z_2 = 0.1z_1$, the right panel of Figure 7 shows the evolution of the output samples generated by both generators for this setup. As shown in this experiment, two generators cannot learn the distribution of digits 8 and 2. While we do not currently have a mathematical argument for this observation, we conjecture that the hidden space ($z$-space) is one of the essential pieces in the demixing performance of the proposed demixing-GAN. We think that having (random) independent or close orthogonal vector $z$'s for the input of each generator is a necessary condition for the success of learning the distribution of the constituent components, and consequently demixing of them. Further investigation of this line of study is indeed an interesting research direction, and we defer it for future research.

In addition to the hidden space, here we design some experiments in the generator space that reveals the condition under which the demixing is failed. In particular, we consider the airplane images in Quick-Draw dataset. To construct the input mixed images, we consider randomly chosen images of

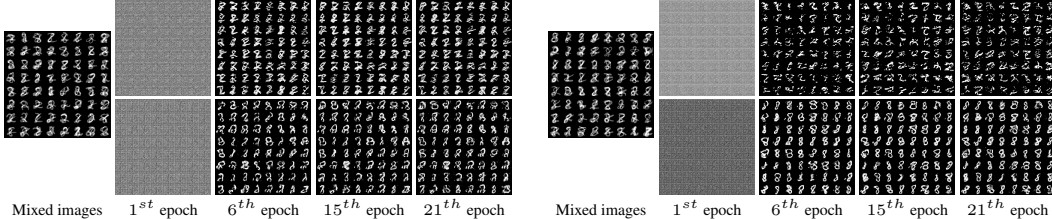

Figure 7: Failure of the demixing-GAN for demixing of digits 2 and 8 from their superposition (exploring in $z$-space). Left Panel: evolution of output samples by two generators for $z_1 = z_2$. Right Panel: evolution of output samples by two generators for $z_1 = 0.1z_2$.

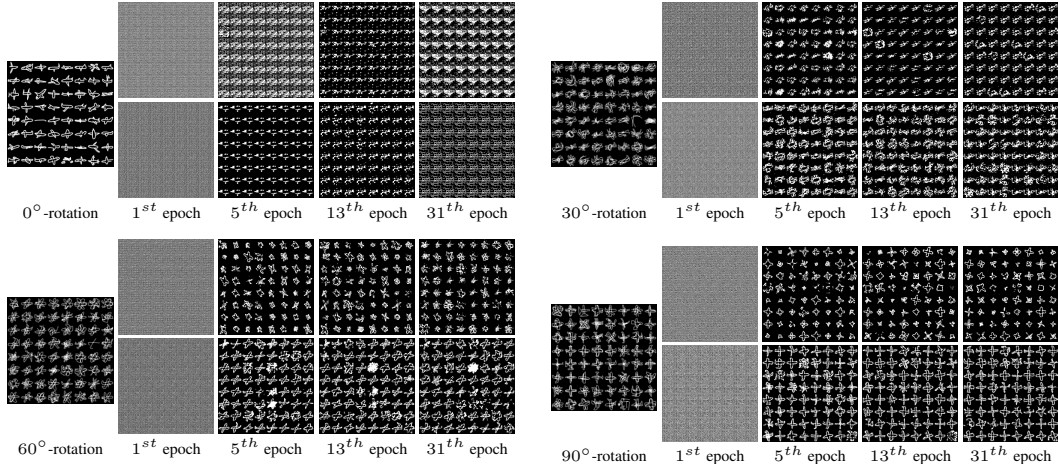

Figure 8: Failure of the demixing-GAN (exploring in the generator space). Evolution of output samples by two generators. Top Left: Mixture of two $0°$ rotated images. Top Right: Mixture of $0°$ and $30°$ rotated images. Bottom left: Mixture of $0°$ and $60°$ rotated images. Top Right: Mixture of $0°$ and $90°$ rotated images.

the airplane from 16000 images as the first component. Then, the second component is constructed by rotating exactly the same one in the first components in a counterclockwise direction. We consider 4 different rotations, $0°$, $30°$, $60°$, $90°$. This experiment is sort of similar to the one in the right panel of Figure 6 in which we have seen that demixing-GAN can capture the internal structure in the airplane dataset by clustering them into two types. Now we perform the demixing-GAN on these datasets. Figure 8 illustrated the the evolution of the generators for various rotation degrees. The top left panel shows the case in which exactly both components are the same. Obviously, the demixing, in this case, is impossible as there is no hope to distinguish the components from each other. Moving on, in the other panels of Figure 8, we have different rotation settings. As we can see, once we move forward to the $90°$, both generators can capture the samples from the airplane distribution; however, as not clear as the case in which we had added the airplane shapes randomly to construct the input mixed images. We conjecture that changing the orientation of one component can make it incoherent to some extent from the other component, and consequently makes the demixing possible. In other words, we see again when two images show some distinguishable structures (in this case, the first one has 0-oriented object and the other is the same object, but rotated $90°$ counterclockwise), then the demixing-GAN can capture these structures.

## 5 CONCLUSION

In this paper, we considered a GAN framework for learning the structure of the constituent components in a superposition observation model. We empirically showed that it is possible to implicitly learn the underlying distribution of each component and use them in the downstream task such as demixing of a test mixed image. We also investigated the conditions under which the proposed demixing framework fails through extensive experimental simulations and provided some theoretical insights.

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
