# OpenReview forum: "Unsupervised Demixing of Structured Signals from Their Superposition Using GANs"
_ICLR.cc/2019/Workshop/DeepGenStruct — DeepGenStruct 2019_

### Official Review · AnonReviewer1 · 2019-04-14
**Interesting initial investigation in the GANs for demixing.**

**Rating:** 4
**Confidence:** 2

**Review:**

The authors investigate the use of a structured generative model
to perform demixing of data in an unsupervised way.
The paper is well written (clearly highlights the lack of supervision prevents training a conditional generative model, and how structure is here key for both learning the model and performing demixing at inference time), and has thorough experiments on simple datasets.
The main limitation - but one which the author recognize and start to investigate  - is that there are no guarantee the current structure is an inductive bias strong enough  to guarantee that the recovered separated signals correspond to the desired ones.
I worry that for complex datasets the approach would not yield the desired results.
Also - one could argue the archetypal problem for source separation is the cocktail party problem - it would have been interesting to try some audio data.

---

### Official Review · AnonReviewer2 · 2019-04-16
**A simple method for linear demixing of signals with GANs**

**Rating:** 3
**Confidence:** 3

**Review:**

The authors present a simple method for training GANs to demix images. They train two separate generator networks, take the sum of their output, and then perform inference by finding the latent vectors for both GANs which minimize the distance in pixel space to the original (mixed) image, with a regularization penalty on the magnitude of the latent vector.

While the approach is very simple, the experiments looked promising. When the two data distributions were very different, the demixing GAN was able to cleanly separate out the two classes. I would have appreciated more rigorous comparison against alternative methods, especially more on ICA and possibly NMF as baselines. More comparison on real world data would be interesting as well - for instance quite a lot of biomedical data contains mixed signals from different types of tissue and automatic demixing is very valuable there.

---

### Decision · Program_Chairs · 2019-04-19
**Acceptance Decision**

**Decision:**

Accept

**Comment:**

Accepted